# Design of an FPGA-Based Controller for Fast Scanning Probe Microscopy

**DOI:** 10.3390/s24186108

**Published:** 2024-09-21

**Authors:** Leonardo Gregorat, Marco Cautero, Sergio Carrato, Dario Giuressi, Mirco Panighel, Giuseppe Cautero, Friedrich Esch

**Affiliations:** 1DIA (Dipartimento di Ingegneria e Architettura), University of Trieste, 34127 Trieste, Italy; carrato@units.it; 2Elettra-Sincrotrone Trieste S.C.p.A. Science Park, Strada Statale 14, km 163.5, 34149 Basovizza, Italy; marco.cautero@units.it (M.C.); dario.giuressi@elettra.eu (D.G.); giuseppe.cautero@elettra.eu (G.C.); 3Dipartimento di Fisica, University of Trieste, 34127 Trieste, Italy; 4Laboratorio TASC, CNR-IOM (Istituto Officina dei Materiali), Strada Statale 14, km 163.5, 34149 Basovizza, Italy; mirco.panighel@uni.lu; 5INFN (Istituto Nazionale di Fisica Nucleare), Sez. di Trieste, Via Valerio 2, 34127 Trieste, Italy; 6Chair of Physical Chemistry and Catalysis Research Center, Department of Chemistry, TUM School of Natural Sciences, Technical University of Munich, Lichtenbergstr. 4, 85748 Garching, Germany; friedrich.esch@tum.de

**Keywords:** scanning probe microscopy, field programmable gate array, digital signal processing, fast scanning tunneling microscopy, atom tracking

## Abstract

Atomic-scale imaging using scanning probe microscopy is a pivotal method for investigating the morphology and physico-chemical properties of nanostructured surfaces. Time resolution represents a significant limitation of this technique, as typical image acquisition times are on the order of several seconds or even a few minutes, while dynamic processes—such as surface restructuring or particle sintering, to be observed upon external stimuli such as changes in gas atmosphere or electrochemical potential—often occur within timescales shorter than a second. In this article, we present a fully redesigned field programmable gate array (FPGA)-based instrument that can be integrated into most commercially available standard scanning probe microscopes. This instrument not only significantly accelerates the acquisition of atomic-scale images by orders of magnitude but also enables the tracking of moving features such as adatoms, vacancies, or clusters across the surface (“atom tracking”) due to the parallel execution of sophisticated control and acquisition algorithms and the fast exchange of data with an external processor. Each of these measurement modes requires a complex series of operations within the FPGA that are explained in detail.

## 1. Introduction

Scanning probe microscopes (SPMs), such as STMs (scanning tunneling microscopes) and AFMs (atomic force microscopes), stand at the forefront of atomic-scale resolving techniques in surface analysis, providing unrivaled information on surface topography and physico-chemical properties. As pivotal tools in nanotechnology and materials science, these instruments continue to push the boundaries of resolution, enabling scientists to explore and manipulate surfaces and supported nanoparticles [1].

SPMs employ a tip or a cantilever as a local probe to study the sample surface. As visible in Figure 1, the probe–sample interaction changes between different techniques, e.g., from the tunneling current in STMs to attractive or repulsive forces or force gradients in AFMs [2]. Due to the diverse interactions and differences in the experimental setups, dedicated electronic controllers and readout schemes are required. SPMs achieve imaging through a raster scan of the probe along the surface. While SPMs can reach a high spatial resolution, the scanning process tends to make them inherently slow, typically requiring several seconds or minutes to acquire images [3].

The limited SPM image acquisition speed is a critical limit of these instruments. This often precludes the tracking of interesting surface dynamics that arise when a system is driven out of equilibrium, e.g., upon external stimuli such as changes in gas atmosphere or electrochemical potential that often occur within timescales shorter than a second [4]. These phenomena include chemical reactions, surface diffusion, and self-assembly, which, for example, cannot be slowed down by lowering the temperature when critical temperatures must be exceeded to allow surface dynamics to occur.

The main challenges in FastSPM consist primarily in the limited bandwidth of the preamplifier electronics and the unwanted excitation of mechanical resonances of the scanning probe. However, recent years have seen a substantial improvement in the scan speed of modern SPMs, achieved through both enhancements in electronic control systems and design strategies aimed at avoiding mechanical resonances [4,5,6,7,8]; for example, by building dedicated instruments with high resonance frequency scanners [9,10,11], with separated scan heads [12,13], or implementing specific scanner shapes and probe trajectories [14,15].

This advancement has facilitated the measurement of various previously inaccessible dynamic processes such as the on-surface synthesis of polymers [16], the intricate adsorbate diffusion in dense coadsorbate layers [17], phase-transition in surface reconstruction [18], electrode surface dynamics at the atomic scale [19], or the details of graphene growth [20,21].

The factors that limit the acquisition speed of a standard SPM are given by: (i) the signal detection speed, (ii) the mechanical stability of the SPM, and (iii) dedicated control electronics that handle high data throughput and smooth scanning motion protocols. While a fast sensing device, e.g., a fast preamplifier in an STM, can adequately increase the detection speed, the construction of dedicated SPMs with high eigenfrequencies for fast operation under full vertical feedback is much more demanding. While several impressive examples have been described in the recent literature [4,9,19,22], these dedicated instruments are typically tailored for specific applications and are demanding in cost and operation, making them less accessible for the general SPM community. Alternatively, one can drive the SPM in a quasi-constant-height motion with sinusoidal fast scan movements at frequencies that do not excite the eigenfrequencies of the microscopes; a condition that can astonishingly often be met [23]. This allows us to overcome the mechanical limitations of commercially available SPMs.

Here, we focus on the third limitation to be overcome: the required fast control electronics. For this purpose, we propose an add-on system that enables precise and fast motion of the probe of any standard commercial SPM while remaining completely transparent for the standard SPM controller [24]. This FastSPM system converts most standard SPMs into fast scanning-capable devices without hardware modifications. Previous versions of the system were based on the National Instrument’s PXI crate housing an FPGA unit (PXI-7951) and a RF input/output unit (NI-5781) [24]. This approach simplified eventual modifications by non-experts in FPGA programming but severely restricted the programming opportunities due to hardware and high-level LabVIEW-FPGA-language limitations. Nevertheless, this system reliably achieved high-speed (up to video-rate) image acquisition performance in different standard commercial SPM setups without any requirement concerning microscope or control hardware customization [20,25].

This article introduces a redesigned and improved transparent add-on system for commercial SPMs, now featuring a novel FPGA-based architecture. This new FastSPM system includes not only fast imaging functionality but also atom tracking capabilites to track the movement of individual surface features. Moreover, it provides useful tools, such as error topography and resonance characterization, to improve its usability. The architecture, system integration, and various measurement modes are described in Section 2, highlighting their experimental requirements and firmware implementation. Section 3 demonstrates the system’s reliable performance in diverse environments—ranging from ultra-high vacuum to ambient pressure and electrolyte—using different SPM methods (STM and contact-AFM) and standard controllers. In conclusion, Section 4 provides a thorough discussion of the results obtained in this work, including a comparison with alternative state-of-the-art solutions.

## 2. FastSPM System Architecture

The FastSPM system presented in this work is composed of two units: a control unit (CU) and a high-voltage unit (HVU). The FPGA-based CU coordinates the probe movement and data acquisition, and it implements custom functionalities. The HVU amplifies the CU probe movement signals and adds them to the high-voltage offset signals delivered by the standard SPM controller. The x-, y- and z-signals thus obtained by the HVU drive the piezoelectric actuators of the SPM probe, e.g., piezoelectric tubes [26]. Figure 2 provides a schematic overview of how the CU and HVU interface with each other and integrate into a standard SPM system setup. This demonstrates the modularity of the FastSPM system, allowing each unit to be independently upgraded or modified (e.g., adding new tools to the CU or adjusting the HVU for different voltage ranges), as long as their core functionalities and interface remain compatible. Note that Figure 2 presents a logic diagram of the connections. The x- and y-signals generated by the standard SPM controller, and the ones applied to the piezo tube, are actually differential. For this purpose, the HVU converts the additional x- and y-signals generated by the CU to differential ones.

The CU is equipped with two high-speed signal inputs, the connection of which depends on the used SPM technique. One input measures the probe–sample interaction signal (tunneling current for STMs or, e.g., error signal for AFMs) and is mandatory, while the other input is optional and may be connected to the z-monitor (or the z-sensor signal) of the standard SPM controller. This signal is used for atom tracking measurements and error topography, as described in Section 2.2.3.

While the FastSPM architecture in Figure 2 is very similar to the previous version [24], it presents a new unified wiring and firmware scheme that allows for a seamless transition between the various measurement modes (fast imaging and atom tracking) without the need to change wiring connections.

To enable the various measurement modes, the CU generates two sets of signals for both the x- and y-axes. The first set are the control signals required for fast imaging, for atom tracking (circular probe movement), and for resonance characterization—in the following denoted as “fast signals”. The second set does not require high speeds but enables slow scans of the sample, and in particular, continuous background movements that are pivotal for compensating for mechanical drifts in fast imaging or high-speed measurements, to generate the probe movement for the tracking of diffusing particles and to generate standard probe movements that can be used for calibration purposes. These signals are referred to as “slow signals” below.

Using the HVU as an adder for the signals generated by the CU allows it to work in parallel and in synergy with the standard SPM controller. When the CU is not in use, the system is completely transparent for the standard SPM electronics and can thus be operated by the standard SPM controller as usual. This allows one to seamlessly and instantly switch between standard (slow) and fast image or tracking acquisition modes in the same spot without altering the probe–sample contact, as the probe does not need to be retracted.

### 2.1. FastSPM Hardware Architecture

The FastSPM’s CU and HVU are housed in two independent units with two separate power supplies. This separation improves the deployment in existing standard SPMs and minimizes electronic interference between the different components.

The HVU is a fully analog electronic system which superimposes, for each axis of motion (x, y, z), the signals generated by the CU (both slow and fast) to the HV signals delivered by the standard SPM controller that represent the lateral offset around which the fast SPM measurements are performed. Since the HV signals are typically differential for the x- and y-axes and single-ended for z, the HVU is based on five APEX PA90 operational amplifiers—one for the z-axis and two for x- and y-axes, respectively—and has an input and output range of ±150 V with a bandwidth of 20 kHz. This makes the FastSPM compatible with most SPM controllers and piezoelectric actuators. The HVU independently amplifies the CU signals with a manually selectable gain for each channel before summing them to form the HV signals.

The CU is an FPGA-based electronic unit that implements the various measurement modes by generating the control signals and acquiring the signals in a synchronized manner. The CU hardware has been designed using a modular approach to allow for easier upgradeability and customization if required. For this reason, the CU functionality has been divided into three different printed circuit boards (PCBs). A picture of the inside of the CU is shown in Figure 3a, with the different PCBs highlighted.

The main PCB, highlighted in red in Figure 3a, contains an Altera Cyclone^®^ V FPGA (5CGXFC5C6F27C7), two small form-factor pluggable (SFP) connectors for the communication with a host personal computer (PC) via Gigabit Ethernet, some digital I/Os, and an FPGA mezzanine card (FMC) connector. The board has been developed in-house as a general-purpose PCB, not limited to the FastSPM system.

A second board, the FMC PCB, shown in blue in Figure 3a, was designed specifically for the FastSPM application. This board is equipped with three dual-channel, 14-bit, 1.389 MS/s digital-to-analog converters (DACs) (DAC8802), which generate the five signals used by the HVU for probe positioning. A dual 16-bit 100 MS/s analog-to-digital converter (ADC) (AD9655) is used for acquisition. There are also two slow 18-bit ADCs (ADS8691) used to log additional data during the experiments, such as current setpoint and probe bias, and a K-type thermocouple input (MAX31855K) to monitor the sample temperature. A rendering of this PCB with the main components highlighted is shown in Figure 3b.

The third PCB, called the conditioning PCB, shown in orange in Figure 3a, is used to interface the FMC board with the HVU and to manage the input signals, amplifying them if necessary and protecting the critical components on the FMC board. As they serve different purposes, the slow x- and y-signals are amplified here to cover a range of ±10 V, while the fast x, y, and z output signals cover a range of ±2.5 V before being summed in the HVU. The factor 4 between the two ranges can also be used to effectively extend the resolution of the slow signal from 14 to 16 bits: if the HVU gain is the same for slow and fast signals, the two least-significant bits (LSBs) of the fast signals are used as the two LSBs of the slow signal. The fast input signals have an overvoltage protection system and a digitally selectable input range of ±1 V or ±10 V to suit a wider range of SPMs and external amplifiers. In addition, the conditioning PCB allows for the use of an LVCMOS-compatible signal to trigger the start of a fast imaging or atom tracking measurement, allowing for synchronization with external instruments.

### 2.2. FastSPM FPGA Firmware

The Altera Cyclone^®^ V FPGA on the main PCB orchestrates all the operations required for the various functionalities implemented. To reduce the FPGA area usage, all the functionalities share as much logic as possible. This also allows for a seamless transition between different measurement modes, which was not possible in the previous FastSPM system. The general structure of the FPGA firmware is represented in Figure 4.

The “network wrapper” module implements a four-port UDP protocol over a Gigabit Ethernet and is used to connect the system to a host PC that controls the FPGA via a LabVIEW-based software. The four UDP ports, between 2047 and 2050, are used for different logical tasks: port 2047 is the command port and is used to configure the instrument using 4- or 8-byte commands. Port 2048 is the fast data port and is used to write the fast scan waveform to the FPGA and to send the measured fast signals to the PC. Port 2049 is an auxiliary data port and it is used to log the auxiliary data channels such as bias, setpoint, and temperature to the PC. Finally, port 2050 is the x/y data port and is used in the atom tracking and the error topography modes to provide the feature position or the x- and y-tracking errors to the host PC.

#### 2.2.1. Fast Imaging Measurement Mode

The main measurement mode of the FastSPM is fast imaging. This is used to obtain an image of the sample surface, as with the standard SPM controller, but with increased time resolution. Typically, SPMs operate with a z-feedback that keeps the probe–sample interaction constant and uses the z-signal to generate the image as it scans along x- and y-axes [27]. With this constant input signal scheme, the acquisition speed is typically limited by the bandwidth of the control loop and the mechanical stability of the SPM. Modern fast SPMs address this problem by improving the mechanical properties of the probe and/or moving stages, allowing for faster closed-loop operation while avoiding resonances.

In order to enable fast imaging on standard SPMs where it is not possible to improve the mechanical properties, the FastSPM system avoids scanning motions with high-frequency components that would excite mechanical resonances by applying a pure sinusoidal motion on the fast scan axis (typically one kHz or more) while maintaining a triangular motion on the slow scan axis (typically a few Hz). The fast scan sine frequency can be tuned to avoid mechanical resonances but will result in distorted images that can later be corrected by Delaunay triangulation and interpolation [28].

The limited mechanical stability of standard SPMs also precludes the use of a fast vertical feedback loop while capturing fast images. Therefore, the FastSPM system operates in a quasi-constant-height mode with a feedforward approach [29]: while residual sample tilt in the slow scan direction can be compensated for by the vertical feedback of the standard controller, tilt in the fast scan direction can be compensated for by applying a sinusoidal z-offset signal of an appropriate amplitude and phase. This compensation can be avoided if the fast scan movement follows the contour lines of the tilted surface by rotating the scan directions as described below.

The waveform for the fast scan movement—the same as that for the z-compensation signal—is stored in an 8192 × 16 bit dual-port on-chip RAM inside the FPGA. The samples are written from the host PC through the network wrapper and are read by two equal but independent finite state machines (FSMs) that make up the “sin-controller” HDL modules in Figure 4. Each FSM reads the waveform sample-by-sample at a frequency of ∼1.389 MHz (50 MHz / 36) to match the timing of the DACs, allowing for a multiplexed RAM reading scheme. For example, to obtain a 1 kHz signal on the x- and z-outputs, the stored waveform occupies only 1389 words of the 8192 available—calculated by dividing the DAC sample rate by the desired output frequency. This approach was chosen to achieve a deterministic period-by-period motion along the fast scan axis with the ability to generate further arbitrary movements. The sin-controllers also scale the signal stored in the RAM to obtain the desired output voltages. During start and stop stages of the fast image acquisition, the sin-controller FSM slowly increments and decrements the scan amplitudes at the fast scan frequency to avoid high-frequency contributions from abrupt changes in the probe motion that could cause instability. These gradual changes occur at the waveform zero crossings to avoid discontinuities. Similarly, the sin-controller also handles signal phase changes by reading the RAM at a shifted address. Runtime phase changes are then applied gradually at waveform maxima, again to avoid discontinuities. Runtime control of amplitude and phase is required to efficiently tune and scale the feedforward z-compensation signal according to the transfer function of the piezoelectric actuator at a given scan frequency.

The triangular motion in the slow scan direction is generated at runtime by another HDL module, the “tri-controller”. Here, an FSM increments and decrements a register by a fixed value, set by the host PC, to obtain the desired amplitude. In addition, the tri-controller can progressively reduce the increment/decrement constant to smooth the transition as a change of direction is approached. This feature helps to further reduce the contribution of higher-order harmonics due to the reversal of the slow probe movement. This is achieved at the cost of slight distortion at the top and bottom of the image, which can be corrected in the post-processing stage. Synchronization of the tri-controller and the sin-controllers is achieved by design, using a sequential start procedure controlled by the network wrapper. In this sequential start, the network wrapper initiates the sin-controllers, which, upon reaching the steady-state, starts the tri-controller. The tri-controller ultimately triggers the acquisition at the first line of an image. Figure 5a shows a time domain acquisition of an example start procedure, with the fast scan signal in blue and the slow scan signal in orange.

As an alternative to z-compensation, the scan directions can be rotated in order to follow the surface contour lines with the fast scan motion as mentioned above. In this way, the slow scan follows the sample tilt gradient in a way that it can be accounted for by the vertical feedback of the standard SPM controller. To do this, the signals generated by one of the sin-controllers and the tri-controller are multiplied by a configurable rotation matrix pre-calculated by the host PC. The matrix multiplication uses four DSP multipliers and allows for arbitrary scan directions with respect to the physical x- and y-axes of the piezoelectric actuator, greatly simplifying the sample tilt-compensation. An example scan path is shown in Figure 5b, measured using an oscilloscope for a rotation angle of 20°, with the tri-controller set to smoothed slow scan reversal.

Once the fast scan movement has reached a steady state, the ADC data of the probe–sample interaction signal is averaged by a factor of 4 to reach a data rate of 25 MS/s. The averaged data are sent to the network wrapper via a dual-clock FIFO (first in first out) to ensure a proper clock domain crossing (CDC) between the network interface clock domain and the control logic one. This is needed to accommodate the different clocks required by DACs/ADCs and the SFP, avoiding metastability [30]. The FSM responsible for UDP port 2048 sends these samples to the PC in 4000-byte packets using jumbo frames to reduce overhead. Each UDP packet has a 4-byte header, indicating the scan line to which the sample belongs and an increment for reordering. While FPGA-based filtering and image reconstruction would be possible, the system uses the available bandwidth to delegate the image processing task to the host PC, which has greater flexibility, as well as further data analysis, e.g., via FFT.

During fast imaging, the “offset logic” module—that is, the module responsible for actuating the slow movements of the probe along the sample surface—is controlled directly by the host PC via dedicated commands that drive the slow x and y DACs. In this measurement mode, it is possible to use these signals to explore the sample surface by incrementally changing the lateral offset and also to compensate for slow background drift, which the host PC quantifies via image cross-correlation. The ability to induce offsets of a controlled size is also used to calibrate the image size of the fast movies by registering the resulting image shifts. This feature is required because the piezoelectric actuators of SPMs driven at high scanning frequencies, close to the mechanical eigenfrequencies of the instrument, no longer operate with a transmission factor of 1. In order to generate calibrated slow motion, a “test pattern” module has been added. This module implements an FSM that generates a square movement of programmable size and speed along the x- and y-axes, similar to the generation of the triangular movement, but in two dimensions.

#### 2.2.2. Atom Tracking Measurement Mode

The atom tracking mode is used to track individual features of interest as they move across the surface being investigated, providing their position with sub-atomic spatial resolution and sub-millisecond time resolution. This is achieved via double lateral positioning feedback in the x- and y-directions that locks the scanner probe onto the feature to be tracked (protrusions or depressions) [31]. The basic idea is to rotate the probe around the feature of interest, such as an atom or a nanoparticle, while monitoring the probe–sample interaction signal. If the atom is centered under the probe, the signal is constant along the circular path. Otherwise, if the feature is off-center, the signal will vary, with a phase that depends on the direction of the offset, indicating a position error that can be corrected by a lateral PI (proportional-integral) controller. The error signals along the x- and y-axes can be easily obtained as the x- and y-components of a lock-in demodulator, also known as the in-phase and quadrature (I/Q) components. The lock-in amplifier detects the amplitude and phase of small periodic signals with a high signal-to-noise ratio based on a reference signal with which the signal to be analyzed is modulated [32]. The first stage of the lock-in amplifier is an I/Q mixer in which the input signal is simultaneously multiplied by the reference signal and its 90° phase-shifted copy. Given an input signal with amplitude *A* and phase φ, $V_{in}=A\cos\left(\omega t+\varphi \right)$, and the reference signal $V_{ref}=2\cos\left(\omega t+\vartheta \right)$, where ϑ is the selectable lock-in phase, mixing results in the following in-phase, $V_{I}$, and quadrature, $V_{Q}$, components:(1)$$V_{I}=V_{in}\cdot V_{ref}=A\cos\left(\omega t+\varphi \right)2\cos\left(\omega t+\vartheta \right)=A\cos\left(\varphi \;-\;\vartheta \right)+A\cos\left(2\omega t\;+\varphi +\vartheta \right)$$
(2)$$V_{Q}=V_{in}\cdot V_{ref}^{90^{\circ }}=A\cos\left(\omega t+\varphi \right)\left(-2\sin\left(\omega t+\vartheta \right)\right)=A\sin\left(\varphi \;-\;\vartheta \right)-A\sin\left(2\omega t\;+\varphi +\vartheta \right).$$

Ideally, low-pass filtering of these signals will remove any component at 2ω, leaving only the DC signals:(3)$$V_{IF}=A\cos\left(\varphi \;-\;\vartheta \right)\;\;\;\;\;and\;\;\;\;V_{QF}=A\sin\left(\varphi \;-\;\vartheta \right),$$
whose values depend on the input signal amplitude and the difference between the input and lock-in phases. $V_{IF}$ and $V_{QF}$ can be easily identified as the real and imaginary terms of the complex vector $Ae^{j\left(\varphi -\vartheta \right)}$. Thus, it is possible to obtain the amplitude and phase of the input signal, or, as in our case, the signal component along the real and imaginary axes, which can be aligned with the physical x- and y-axes by tuning the lock-in phase. A simulated circular motion on an off-center protruding feature is shown in red in Figure 6a. This motion is obtained by a cosinusoid along the x-axis and an inverted sinusoid on the y-axis, as shown in Figure 6b (blue and orange curves, respectively). Here, the additional green curve indicates a normalized interaction signal (assumed to be linear with the protrusion height) obtained during circular motion. The I/Q mixing signals, obtained by multiplying the interaction signal by the x-axis and y-axis signals, respectively, are shown in Figure 6c. Low-pass filtering of the signal—here obtained by averaging the mixed signals—results in the dashed lines. As expected for the position offset shown in the example in Figure 6a, the average of the quadrature signal, corresponding to the error in the y-direction, exceeds that of the in-phase signal, corresponding to the error in the x-direction.

To keep the circular movement centered on the feature being investigated, x- and y-errors are used by two closed-loop PI controllers, similar to that typically used for the vertical feedback in standard SPMs [33]. The output of a PI controller is the sum of two terms: a proportional term P applies a correction proportional to the current error with a coefficient $K_{P}$, the value of which results in a slower or faster response. As the P term is not able to completely cancel out the steady-state error, a further integral term I is used. This I-term is proportional to the integral of the error with a coefficient $K_{I}$, allows the steady state error to be eliminated, and is more robust to noisy inputs than the P-term [33].

As the typical preamplifier bandwidths of around 10 to 200 kHz are used in conjunction with the FastSPM system, the circular motion of the probe typically occurs at a rotation frequency of 1 to 10 kHz and is generated by the two sin-controllers. The sin-controllers use the waveform stored in the on-chip RAM to output the same signal but with a programmable phase difference between the x and y fast DACs, typically 90° or –90°. The amplitude of the circular motion can also be controlled by the sin-controllers, independently for the two axes, to account for possible different transfer functions of the piezoelectric actuators. The circular motion’s diameter must be slightly larger than the features of interest, and its dimensions must be chosen with care: a larger circle allows fast moving features to be tracked reliably with less spatial accuracy, while a narrower circle is very sensitive to fine motion but is not suitable for fast features that escape the rotating probe.

The probe–sample interaction signal measured by the ADC during the circular movement is processed by a digital lock-in located in the FPGA, which demodulates the signal via multiplication with the sinusoidal excitation waveform stored in the on-chip RAM and a 90° shifted version as reference signals. These signals have a programmable phase relationship with the excitation signals generated by the FastSPM system, allowing the lock-in phase to be experimentally tuned to align the resulting in-phase and quadrature error signals with the x- and y-axes of the scanner. Furthermore, by adding a 180° shift to the lock-in phase, it is possible to track vacancies or recessed features instead of protruding ones [34]. The lock-in phase calibration is discussed in more detail in Section 2.2.3 on the acquisition of error topography images.

The digital lock-in uses two hardware DSP multipliers for mixing, while low-pass filtering is performed without multipliers in two possible implementations: a time-constant mode and a circle-by-circle mode. The time-constant mode low-pass filter (LPF) is a sequence of cascaded integrator-combs (CICs) [35] and RAM-based recursive moving averagers. The order and number of stages used is dynamically controlled by the host PC to achieve a transfer function similar to a first- or second-order LPF with an arbitrary time constant, simulating the behavior of an analog lock-in. The circle-by-circle mode, on the other hand, uses the same RAM-based recursive moving average but with a length equal to the number of samples taken during an integer number of circles, i.e., full rotations.

The circle-by-circle filtering allows for perfect rejection of the unwanted signal at twice the rotation frequency, due to the mixing, and at the rotation frequency, due to the up-conversion of the input DC component. If the waveform stored in the RAM consists of *p* points, the resulting digital frequency of the signal is $\omega_{r}=2\pi /\left(72p\right)$ rad/sample, and the ADC acquires 72p samples during each circle, since the ADCs operate at 100 MS/s and the DACs operate at 100/72 MS/s. Given the moving average transfer function [35]
(4)$$H\left(\omega \right)=\frac{\sin\left(\frac{\omega }{2}L\right)}{L\sin\left(\frac{\omega }{2}\right)}e^{-j\omega \frac{L-1}{2}},$$
with the average interval *L*, we can express L=72p·m, with the integer number of circles *m*. It is easy to see that $H\left(\omega \right)$ vanishes for every $\omega =n\cdot \omega_{r}/m$ with the integer *n*. Thus, averaging the input samples taken over an integer number *m* of circles perfectly filters out all the unwanted frequency components caused by mixing.

The x- and y-lock-in filter outputs are used as error signals by two independent FPGA-based PIs. The PI controllers are implemented in the ideal representation [33], with the integral part obtained via backward Euler transformation [36], resulting in the discrete time transfer function: (5)$$H\left[z\right]=K_{P}\left(1+\frac{T_{C}}{T_{I}}\frac{z}{z\;-\;1}\right)$$
where $T_{C}$ is the digital control time step and $T_{I}$ is the integration time, with $T_{I}=K_{P}/K_{I}$. The resulting FPGA PI implementations require only a multiplier, a multiply–accumulate (MAC) operation, and an adder, with the schematic structure shown in Figure 7.

The PI outputs track the feature of interest by driving the slow-scan DACs through the offset logic. The PI outputs are also sent to the host PC via UDP port 2050 to visualize the feature position in real time in the high-level software. The average interaction signal and the z-monitor signal provided by the standard SPM controller are also sent to the host PC via the same port. Note that in the atom tracking mode, the z-feedback is still controlled by the standard SPM controller, which thus works in synergy with the FastSPM system, keeping the average probe–sample interaction constant.

In atom tracking, the test pattern module can be used to optimize the PI parameters. The control signals that define a test pattern movement are generated independently of the lateral PI controller outputs. Both signals are then summed within the offset logic and sent to the slow DACs, and they can act in an antagonistic manner. When tracking an immobile feature and imposing, e.g., a square test pattern movement that simulates its motion, the compensation thus detected by the lateral PI controllers can be optimized with a high accuracy for high responsiveness while avoiding oscillations.

#### 2.2.3. Error Topography Tool

Prior to tuning the lateral PI feedback, the atom tracking mode requires careful tuning of several parameters for proper error detection via lock-in. To simplify this tuning process, an error topography (ET) imaging tool was implemented. ET provides maps of the lock-in in-phase and quadrature signals, i.e., the x- and y-error, together with a map of the z-monitor signal, i.e., the surface topography. In ET measurements, the probe is in continuous circular motion, as in atom tracking, but without the lateral PI feedback. When performing a slow scan around the feature to be investigated later by atom tracking, the lock-in output is measured, as shown in Figure 8a.

Various time delays in the measurement setup result in an overall non-zero phase shift of the incoming signal relative to the circular movement. This phase shift (referred to hereafter as the lock-in phase ϑ) must be determined in order to align the lock-in errors to the physical scan axes so that the PI controllers can generate appropriate error-compensation movements. Correct alignment of the lock-in phase is shown in the ET I/Q maps in Figure 8b. This example shows an already aligned case for the tracking of a protruding feature: in the in-phase map, the error is positive on the left side of the feature and negative on the right side, as required to obtain a corrective action of the feedback on the x-axis. The same considerations can be made for the quadrature map along the y-axis.

The ET maps, especially the z-topography map, can also be used to evaluate the effective amplitudes and phases of the resulting circular motion. Since the piezoelectric actuators may have different responses in the x- and y-directions when driven at frequencies close to their mechanical eigenfrequencies, resulting in different x- and y-transfer functions [23], the x and y DAC signals may require different amplitudes and a phase other than 90° to achieve the perfect circular motion that is required to obtain reliable error signals. By selecting, as a test, slightly larger amplitudes than those used for precise positioning, the resulting toroidal pattern in the z-topography map can help to obtain a perfect circular symmetry of the motion.

To implement the ET acquisition tool, the FastSPM system works in a similar way to the atom tracking mode, using the sin-controllers for movement and the lock-in for error calculation. The PI controllers are disabled, and the offset logic is driven directly by the host PC step-by-step to implement a snake scan on the slow signal DACs. The lock-in outputs are sent directly to the host PC on UDP port 2050, instead of using the PI outputs, to visualize the error maps in real time. Both amplitude and phase can be adjusted in real time to allow for live tuning of the parameters. Once the optimal parameters have been found, they can be used for the atom tracking measurements, with retuning typically only required when the rotation frequency is changed.

#### 2.2.4. Resonance Characterization Tool

As mentioned above, the measurement modes introduced by the FastSPM system require standard scanning stages to operate at frequencies above the frequency range for which they were originally designed [23], often exceeding the first eigenfrequencies of the piezoelectric actuator. The correct choice of actuation operating frequency, which avoids excitation of resonances, is therefore crucial to obtain stable, undistorted images and reliable error signals. Since a trial-and-error approach to finding optimal operating frequencies is neither simple nor efficient, a resonance characterization tool has been implemented that allows for the identification of the most promising frequency windows for stable operation. The integrated software tool measures the amplitude and phase response of the tunneling current upon oscillatory excitation of an axis of choice (x, y, or z) at distinct frequencies within a given frequency range. Optimal fast-scan imaging frequencies are typically found in those frequency ranges where the amplitude and phase response are constant. This response is best resolved in the most sensitive direction of excitation, i.e., z [23]. Due to the residual tilt of the samples, which translates lateral excitation into signal modulation, the amplitude of the flat frequency response windows, which is proportional to the transfer function, can be used for calibration purposes.

While such resonance characterization could be performed using any external frequency sweep lock-in amplifier, the advantage of a software solution using the lock-in amplifier already implemented within the FastSPM system is that the characterization can be performed in situ, immediately prior to any fast measurement. This is done by using one of the sin-controllers and its waveform memory to generate an excitation signal, which is sent to one of the three fast x-, y-, or z-outputs. To achieve a discrete frequency sweep of the output signal, the host PC continuously updates the waveform stored in the RAM. At the same time, the lock-in integrated in the FPGA demodulates the probe–sample interaction signal and sends the in-phase and quadrature outputs of the lock-in amplifier to the host PC via port 2050, taking into account the lock-in filter settling time for adequate noise filtering combined with responsiveness. The host PC then calculates and displays the measured amplitude and phase values, providing the full frequency response for excitation of the selected axis.

## 3. Experimental Test Measurements

The FastSPM system and its various measurement modes and tools were successfully tested on different standard commercial SPMs (2 STMs, 1 AFM) operating in different environments (UHV, air, electrolyte) throughout the development phase. The only customization required to use the FastSPM on different SPMs is the use of a custom breakout box or cable to interface the standard SPM controller to the HVU and the HVU outputs to the SPM probe.

### 3.1. Fast Imaging

We start with the fast imaging mode to demonstrate the easy and reliable applicability of the FastSPM system on three different experimental platforms, where the frame sequences in Figure 9 have been recorded. The top row sequence was acquired using a UHV setup with an Omicron VT-AFM microscope operated as an STM at room temperature. We observed a static Pt_5_ cluster sitting on a Fe_3_O_4_(001) magnetite surface. Note that the Fe rows of the support are atomically resolved in the 4 frames/s movie with a 100 × 100 pixel resolution. The middle row sequence was instead taken in an electrochemical STM (EC-STM) with a Beetle-type setup [37]. Here, we observe an ordered Pd-octaethylporphyrin monolayer on a Au(111) surface, imaged in a liquid environment, i.e., in a phosphate buffer and electrochemical potential control. Note that stable, undistorted images can be obtained at 12 frames/with a 120 × 120 pixel resolution, resolving the individual molecules. The bottom row adds the example of an AFM experiment taken in air. In this case, a 3 µm pitch and 130 nm height Si-based calibration grating was imaged in contact mode using an Asylum Research/Oxford Instruments MFP-3D microscope. Remarkably, this instrument can be driven over lateral scales in the µm range at 4 frames/s with a 100 × 100 pixel resolution and with no apparent distortion or instability.

### 3.2. Atom Tracking

To demonstrate the fast tracking response that can be achieved using our new FPGA-based system in the atom tracking mode, we report in Figure 10 a test measurement obtained in a UHV-STM on a magnetite-supported Pt_5_ cluster, as shown in the top row of Figure 9. Here, we locked the STM tip onto a cluster at a rotation frequency of 4.74 kHz and measured the lateral PI feedback outputs with a maximum obtainable time resolution of 500 µs while inducing a custom-defined movement. In this case, instead of using the square test pattern of the FastSPM system, we used a more complex treble clef pattern generated by the standard SPM controller, a Matrix system from Scienta Omicron. The movement is defined in a custom matrix-automated task environment (MATE) script, shown in the left track of Figure 10. The arrow indicates the start point and direction of the motion, and the speed is color coded. We then investigated the FastSPM’s ability to compensate for this movement for different PI parameter settings (center and right tracks). To test the limits of the tracking speed, we applied extremely high speeds of 50 nm/s or 100 nm/s in the different clef segments, which are orders of magnitude higher than those of typical surface diffusion processes studied in STM movies [38].

As the experiment shows, atom tracking can indeed keep up with fast-moving objects and reliably measure their speed. In this example, however, 100 nm/s represents the upper speed limit, since tracking instabilities begin to appear in the high-speed tracking segments, especially in the right track where the PI parameters are less optimal.

Another type of instability is indicated by the two red circles in the center track: both discontinuities map a change in shape of the otherwise static cluster. This shifts the detected cluster position first to higher y-values and then back to the original position, resulting in an overall shift of the trace in between these two events. This particular measurement thus demonstrates the accessibility of atomic-scale dynamics via tracking, achieving impressive lateral tracking resolutions down to a fraction of an Å.

## 4. Discussion

The FastSPM system we present in this paper is part of the broader and rapidly growing set of acquisition systems that exploit the enormous potential of FPGAs in many areas of scientific research [39,40,41] by performing sensor data digitalization and processing on them instead of using external instruments.

The presented FastSPM system is used to extend the capabilities of existing SPMs, providing access to the investigation of advanced time-resolved surface dynamics with a easy-to-deploy and flexible approach. The system is based on a modular hardware architecture, modified and improved from an earlier work, consisting of a high-voltage unit and a control unit that works in parallel and in synergy with the existing SPM controllers, using the same probe–sample interaction signal and adding together the scan axis control signals.

The full potential of the FastSPM system is unlocked by the new FPGA firmware that integrates fast imaging and an atom tracking measurement mode and that exchanges data at high speeds with a LabVIEW-based control software. The featured measurement modes allow us to reliably achieve high-speed (up to video-rate) imaging and sub-ms single-particle tracking, as demonstrated, and to seamlessly transition between them. Specifically, the FastSPM system is capable of particle tracking with a 5 µs time resolution and of fast imaging up to 100 × 100 pixels at 200 frames/s (or 200 × 200 pixels @ 100 frames/s), limited by the HVU bandwidth.

The new FPGA firmware described here also includes tools and features to efficiently set up the fast measurements in standard SPMs, which are crucial for successful measurements of highly dynamic systems where in situ runtime optimization is required. An error topography tool allows us to efficiently optimize the lock-in parameters for lateral error determination in atom tracking. A resonance frequency characterization tool, which excites the fast scan motion as in fast movies, allows for easy determination of quiet frequency windows that avoid excitation of the SPM scanner setup. A configurable test pattern movement is used to tune PIs parameters and calibrate the dimensional accuracy. The possibility of arbitrary movements is examined to explore the sample surface and compensate for slow drifts.

State-of-the-art high-speed STMs with 10–100 frames/s can be found in the literature, whether custom-built [8,9,42] or commercially available [43,44]. However, these are based on specific SPM designs. Conversely, we demonstrate here that our FastSTM system can be rapidly applied to various standard SPMs without hardware modification, and it allows for acquisition times to be reduced by at least two orders of magnitude. The achievable imaging frequencies depend on the design of the commercial SPM used. Even though the experimental results presented are slower than 20 frames/s, the system has been shown to be capable of >50 frames/s acquisitions, even in the previous version [20].

With respect to the atom tracking technique, applications for STMs have been known since the 1990 [31,34], but available implementations are, to our knowledge, limited to drift correction applications, if not implemented by slower image-based algorithms [45,46]. This makes our solution, as far as we know, unique in providing reliable detection of surface feature positions with sub-nm spatial resolution and sub-ms temporal resolution.

As experimentally demonstrated, the proposed architecture and available measurement modes, tools, and features allow for easy, reliable, and fast deployment on many commercial SPMs. The novelty of the presented approach does not lie in the individual methods but in their combined implementation and the seamless transition between them. It brings FastSPM measurements to a new level of accessibility for a wide community of users who wish to accelerate their standard SPMs by several orders of magnitude for dynamic measurements. At the heart of this approach is the sophisticated FPGA firmware in the control unit and its simple, reliable integration into standard SPMs via a high-voltage unit.

## 5. Conclusions

This work presents the FastSPM system, an FPGA-based add-on that enhances standard commercial SPMs by enabling high-speed imaging and real-time particle tracking with sub-millisecond resolution. By integrating commercial SPMs with standard control systems, the FastSPM provides a flexible, easy-to-deploy solution that opens up new possibilities for studying fast surface dynamics at the atomic scale.

The FPGA architecture allows for fast imaging and atom tracking measurement modes to be seamlessly combined with optimization tools such as error topography and resonance frequency characterization. This combination overcomes the typical time resolution limitations of standard SPMs, making successful high-speed measurements accessible to a wider range of users. We have demonstrated the use of FastSPM in different environments (from UHV to solid-liquid interfaces) and instruments (from STM to contact AFM).

This work demonstrates how integrating state-of-the-art FPGA architectures into sensors and instruments can unlock previously inaccessible information—in this case, the temporal dimension of fast surface dynamics.

## Figures and Tables

**Figure 1 sensors-24-06108-f001:**
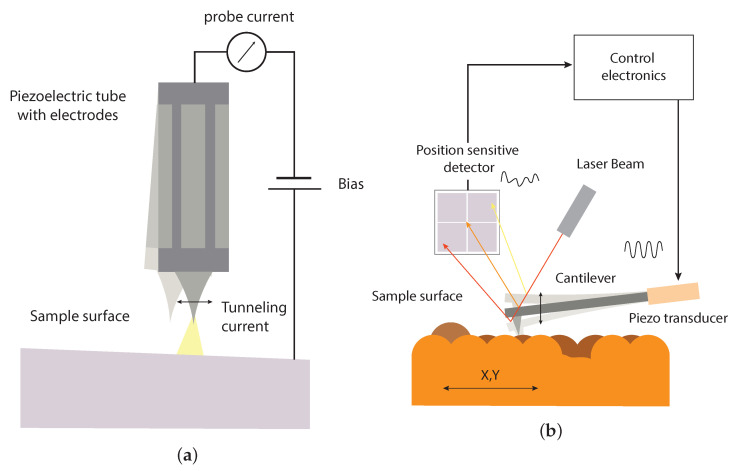
Representation of the most widespread scanning probe microscopes techniques. (**a**) Scanning tunneling microscopy; (**b**) atomic force microscopy.

**Figure 2 sensors-24-06108-f002:**
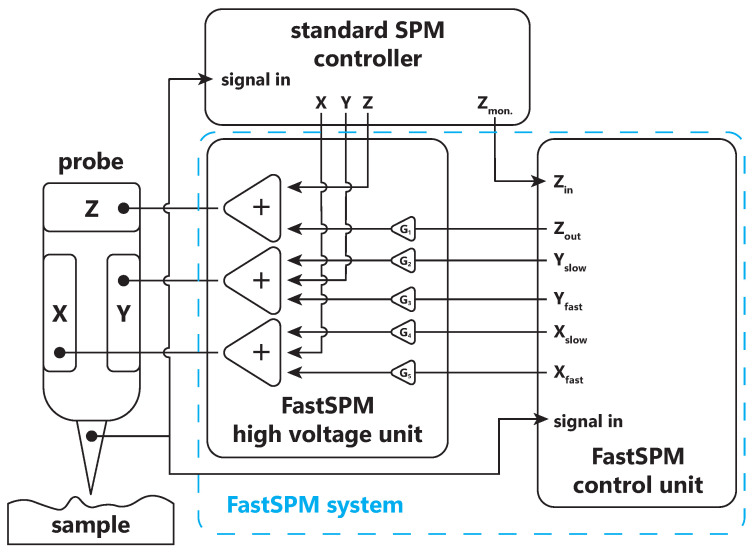
Scheme illustrating the insertion of the FastSPM system (control unit CU and high-voltage unit HVU) between the standard SPM controller and probe (e.g., an STM tip) in the new version of the FastSPM system. The HVU superimposes the CU signals to the standard SPM controller ones before driving the probe actuator.

**Figure 3 sensors-24-06108-f003:**
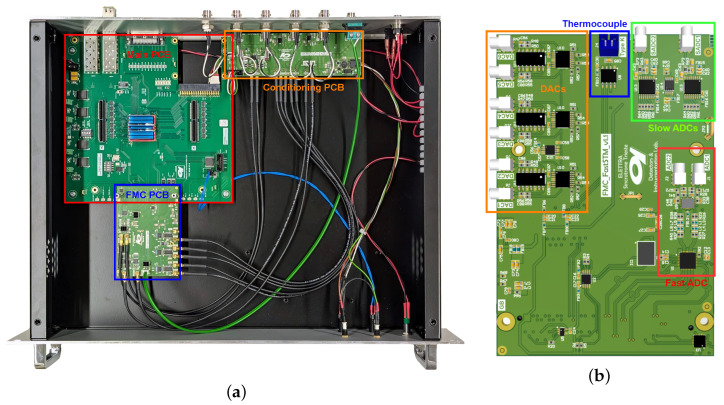
(**a**) Image of the control unit highlighting the 3 PCBs: in red, the main PCB; in blue, the FMC PC;B and in orange, the conditioning PCB. (**b**) Rendering of the FMC PCB with the main blocks framed: in orange, the DACs; in blue, the thermocouple IC; in green, the slow ADCs; and in red, the fast ADC.

**Figure 4 sensors-24-06108-f004:**
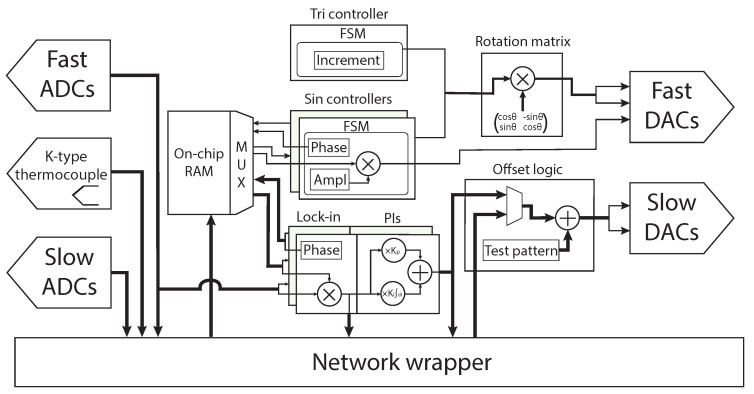
Block diagram of the FPGA firmware structure, presenting the main hardware description language (HDL) modules and data paths.

**Figure 5 sensors-24-06108-f005:**
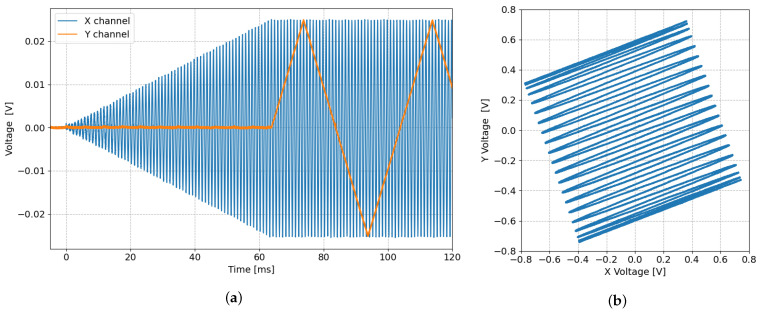
(**a**) Fast (blue) and slow (orange) scan signals generated at the start of an image acquisition. (**b**) Scan path of a fast, rotated image acquisition with smoothed slow scan direction reversal.

**Figure 6 sensors-24-06108-f006:**
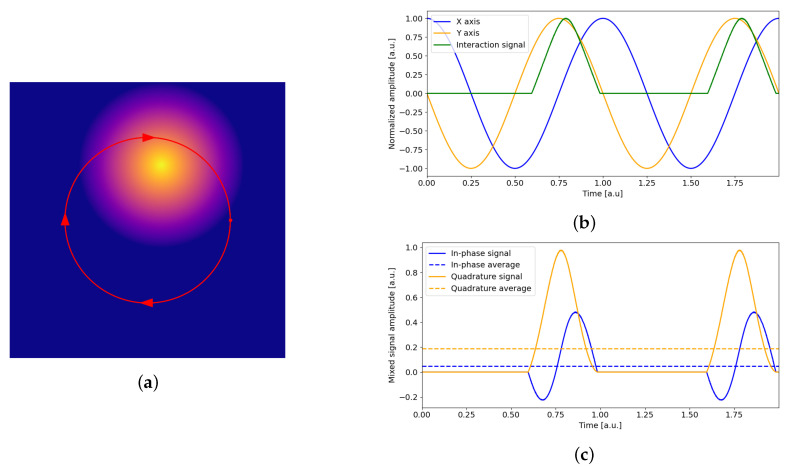
(**a**) Visualization of the circular motion of the probe (red circle) near a protruding surface feature. (**b**) Visualization of the x- (blue) and y-signals (orange) used to obtain the circular motion shown in (**a**) and the resulting interaction signal along the circular path (green), here assumed to be proportional to the feature height. (**c**) Resulting in-phase and quadrature signals and their mean values after demodulation.

**Figure 7 sensors-24-06108-f007:**
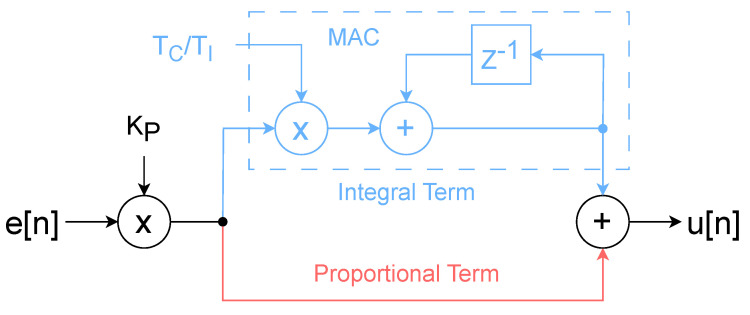
Scheme of the FPGA-implemented PI controller, highlighting the paths of proportional (red) and integral terms (light blue).

**Figure 8 sensors-24-06108-f008:**
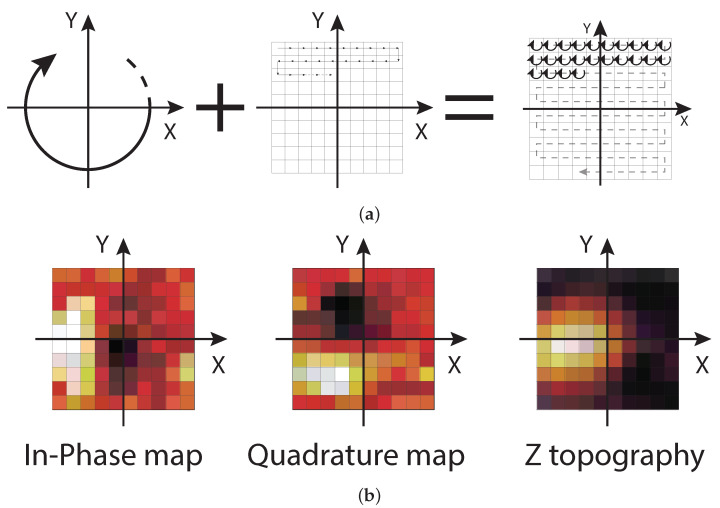
(**a**) Representation of the scan movement used for error topography measurements resulting from the sum of a small, fast, circular movement and a slow conventional scan over the feature under investigation. (**b**) Example of in-phase and quadrature error maps resulting from the lock-in output and of the z-topography, resulting from the z-monitor signal input. Here, the lock-in phase has been correctly set as to align the x- and y-error signals with the physical scan axes (the heatmap indicates negative errors in black, positive errors in white).

**Figure 9 sensors-24-06108-f009:**
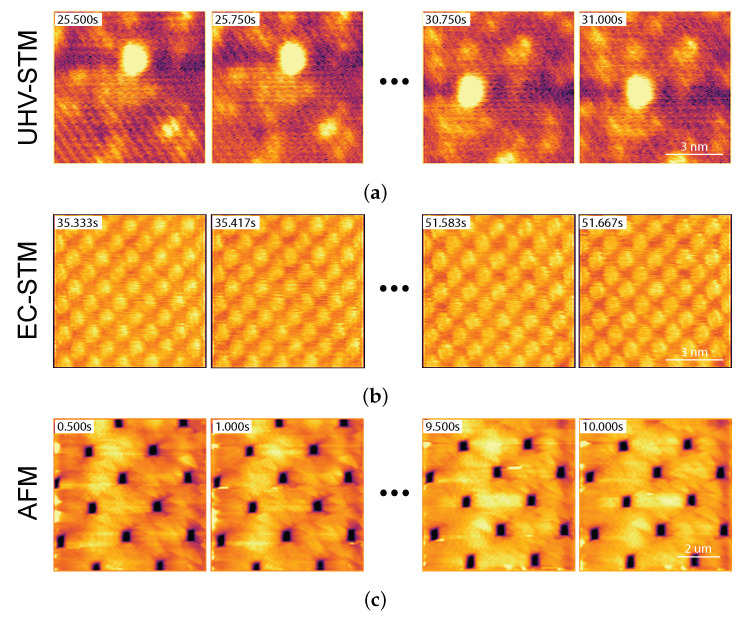
Fast imaging measurements using the FastSPM system: Three sequences demonstrating the applicability of the FastSPM electronics in different standard commercial SPMs and environments: (**a**) Fast STM images of a static Pt_5_ cluster on a Fe_3_O_4_(001) magnetite surface, taken at room temperature in a UHV chamber with an Omicron VT-AFM microscope at 4 frames/s; pixel resolution 100 × 100 pixels; image size 8 × 8 nm^2^. (**b**) Fast STM images of a Pd-octaethylporphyrin monolayer on a Au(111) surface under electrolyte (phosphate buffer, pH = 7, Ar-saturated), taken with a Beetle-type EC-STM at 12 frames/s ($E_{WE}=+0.65$ vs. RHE, $U_{b}=+0.4$ V vs. WE); pixel resolution 120 × 120 pixels; image size 8 × 8 nm^2^. (**c**) Fast AFM images of a Mikromasch TGX1 test grating with a 3 µm pitch and a 130 nm height, taken in contact mode using an Asylum Research/Oxford Instruments MFP-3D microscope (Mikromasch NSC36 probe—0.6 N/m cantilever) at 4 frames/s; pixel resolution 100 × 100 pixels.

**Figure 10 sensors-24-06108-f010:**
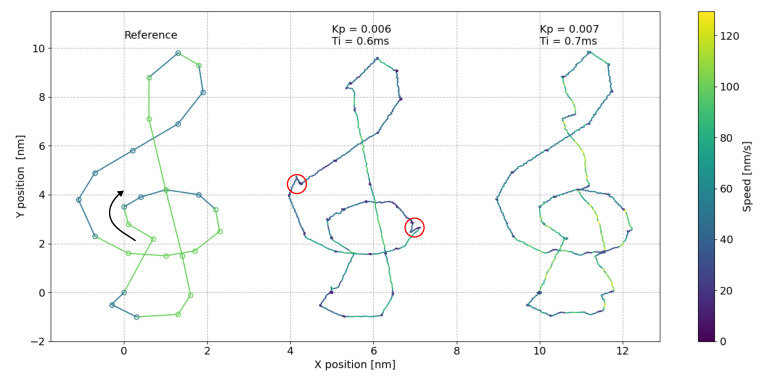
Atom tracking measurement with the FastSPM system in a UHV-STM: (**left**) Treble clef pattern generated by the matrix SPM controller via a MATE script, applied as tip offset. The pattern is defined as linear segments of different speeds (50 and 100 nm/s, color coded); start point and direction are indicated by an arrow. (**center** and **right**) Resulting tracking measurements (lateral PI feedback outputs) with an STM tip locked on a magnetite-supported Pt_5_ cluster at a rotation frequency of 4.74 kHz and measured with a 500 µs time resolution. The two measurements differ in the PI parameter settings, demonstrating the importance of proper parameter tuning for features that propagate at high speeds. Two red circles indicate cluster shape changes that shift the detected cluster position of the trace back and forth by the same amount. The trace between these discontinuities therefore appears to be shifted to higher y-values. With appropriate PI parameters, reliable tracking can be achieved down to a fraction of an Å.

## Data Availability

The data that support this work are available at: https://doi.org/10.5281/zenodo.13744336.

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
