# Peer review of "Design of an FPGA-Based Controller for Fast Scanning Probe Microscopy"

_sensors, 2024, doi:10.3390/s24186108_

Round 1

Reviewer 1 Report

Comments and Suggestions for Authors

The manuscript by Gregorat et al. reports an additional FPGA-based hardware/software to extend the capabilities of standard commercial SPMs, i) enabling fast scanning in quasi-constant height mode, ii) atom tracking. Although the basic idea has already been reported in Ref. [16], there is enough new information on the details of the implementation and some new results which warrant the publication of this manuscript in Sensors.

However, there are several details which make it difficult to understand this manuscript. The authors should address/consider the following questions and suggestions.

1. More details about HVU should be given since it plays an essential role in the system. Compared with the CU, only a few details are given for the HVU. Providing a more detailed block diagram will be useful.  For example, what is the bandwidth of the HVU? Although five APEX PA90 are used, only three amplifiers are shown in Figure 2.

2. L.187 “constant signal scheme’ should read ‘constant input signal scheme’ for clarity.

3. L.209 “HDL” is not defined.

4. L.212 The sentence starting with “For example, …. “, is not clear. How can we get the number, 1389?

5. L.233 “a sequential start procedure” is not clear. Please clarify it.

6. L.249 “to ensure a proper clock domain crossing (CDC)” is not clear. Please clarify.

7. L.256 The term, “offset logic block” should be defined more clearly.

8. L.319 “rotary motion” should read “circular motion”.

9. L.339 and 342 Please include references for “CICs” and “circle-by-circle mode”. Does Ref. [26] explain this?

10. L.369 Explain “test pattern block”. It is not clear.

11. L.406 What is “snake scan”? 

12. Concerning 2.2.4 Resonance characterization tool, consider to include some experimental results that are similar to what are included in Ref. [14].

13. It would be great if the authors can include the actual videos for Figure 9.

14. L.564 Is it fair to claim “reliably achieve video-rate imaging” when they achieved only 12 frames/s?

Author Response

Comment 1: More details about HVU should be given since it plays an essential role in the system. Compared with the CU, only a few details are given for the HVU. Providing a more detailed block diagram will be useful. For example, what is the bandwidth of the HVU? Although five APEX PA90 are used, only three amplifiers are shown in Figure 2.

Response 1: As pointed out by the reviewer, the HVU plays an essential role in the presented system. Even though the operating principle of the HVU is simple to understand, the actual fabrication requires careful selection of the components. When writing this article, given the topic of this special issue, we thought that it was more appropriate to focus on the description of the logic embedded in the FPGA (CU), leaving just the role, specifications and main components of the HVU. Nonetheless, as we recognize that some piece of information may be crucial for a better understanding of our work some details, including the bandwidth, have been added in the main text.

Also, we thank the reviewer for the comment about Figure 2. In the previous caption of this figure, it was pointed out that the signals for the X and Y movements are differentials hence required 2 APEX PA90 each. In order to accommodate for the comments of other reviewers the caption has been shortened, moving these details in the main text in lines 99 and 136.

Comment 2: L.187 “constant signal scheme’ should read ‘constant input signal scheme’ for clarity.

Response 2: We thank the reviewer for the comment, we added “input”, now in line 194, to improve clarity.

Comment 3: L.209 “HDL” is not defined.

Response 3: We thank the reviewer for pointing out the missing definition, which is now included in the first appearance of “HDL”, in the caption of Fig. 4.

Comment 4: L.212 The sentence starting with “For example, …. “, is not clear. How can we get the number, 1389?

Response 4: The DAC has a sample rate of 50/36 MHz (~1.389 MHz), outputting a new sample every 720 ns. On the other hand, a 1 kHz sinusoid has a period of 1 ms. To get the number of samples per period it is possible to divide 1 ms / 720 ns. Without computing the reciprocal of the frequency, it is just possible to divide the sample rate by the desired frequency 1.389 MHz / 1 kHz. This has now been added to the paper in lines 220-221

Comment 5: L.233 “a sequential start procedure” is not clear. Please clarify it.

Response 5: Thanks to the reviewer comment we better detailed the start procedure sequence in lines 242–244.

Comment 6: L.249 “to ensure a proper clock domain crossing (CDC)” is not clear. Please clarify.

Response 6: To better clarify what is clock domain crossing we stressed why it is needed in lines 260- 262 and added reference 30.

Comment 7: L.256 The term, “offset logic block” should be defined more clearly.

Response 7: We thank the reviewer for this observation. As correctly pointed out, the term “block” can be excessively generic and misleading. For this reason, now the manuscript presents the word “module”, which is a more accurate term used in the FPGA lexicon to describe an entity implementing some kind of logic operations. Furthermore, a clearer description of the purpose of this module has been provided.

Comment 8: L.319 “rotary motion” should read “circular motion”.

Response 8: We thank the reviewer for this comment. “Circular” is indeed the correct adjective to describe the movement of the probe and for this reason it has been replaced in the manuscript.

Comment 9: L.339 and 342 Please include references for “CICs” and “circle-by-circle mode”. Does Ref. [26] explain this?

Response 9: While we substituted Ref. [26] with what is now Ref. [35] as it provides more details, Ref. [35] also includes CICs, and is now referenced in line 352 (even though not highlighted). On the other hand, the “circle-by-circle mode” is novel, and we can’t provide an exact reference for it, but it is explained between lines 355–367.

Comment 10: L.369 Explain “test pattern block”. It is not clear.

Response 10: We thank the reviewer for this observation. As reported also in “Response 7”, the term “block” has been replaced by the more appropriate word “module”.

Comment 11: L.406 What is “snake scan”?

Response 11: We thank the reviewer for allowing us to clarify. A “snake scan” is a pattern used to cover a rectangular area, where the probe actively measures during both the forward and backward movements but on different rows, as visible in Fig. 8a in the middle. Citing Wen et al. [10.1109/TCSVT.2011.2106274] “Snake Scan processes the first row from left to right, then the second row from right to left, and then the third row from left to right, and so on”.

Comment 12: Concerning 2.2.4 Resonance characterization tool, consider to include some experimental results that are similar to what are included in Ref. [14].

Response 12: We appreciate the reviewer’s interest in the resonance characterization tool. Given that this manuscript is part of an FPGA-oriented special issue, our focus was primarily on presenting the FastSPM implementation rather than presenting extensive scientific experimental results. The current results were intended to showcase the system's flexibility. We acknowledge the value of the experimental results and plan to include more detailed and scientifically relevant data in future publications.

Comment 13: It would be great if the authors can include the actual videos for Figure 9.

Response 13: The videos will be uploaded in the supplementary information. Furthermore, a link will be provided containing all the measurements presented in this work with both raw and processed data (video/individual frames).

Comment 14: L.564 Is it fair to claim “reliably achieve video-rate imaging” when they achieved only 12 frames/s?

Response 14: We agree with the reviewer’s observation that 12 frames per second does not strictly qualify as video-rate imaging. To address this, we have revised our statement to indicate that the system reliably achieves high-speed imaging up to video-rate when possible, which include sub second imaging. While the presented measurements do not meet the video-rate threshold, due to limitations of the used SPMs, it has the potential for it. The previous version was already capable of video-rate if the SPM hardware permitted it. This is now better explained in lines 533–540.

Reviewer 2 Report

Comments and Suggestions for Authors

In this article, the authors present a fully redesigned FPGA-based instrument that can be integrated into most commercially available, standard scanning probe microscopes. This instrument not only significantly accelerates the acquisition of atomic-scale images, but also enables the tracking of moving features across the surface. Each measurement mode requires a complex series of operations within the FPGA, which is explained in detail. However, there are a few points that could be addressed to further strengthen the manuscript:

1.        In the introduction section, the author should add background information related to the rapid scanning probe microscope, such as the current common methods to improve scanning speed, the existing challenges, etc

2.        The specific advantages of the FastSPM system's fast imaging function should be elaborated.

3.        More comparative experiments should be added to better demonstrate the superiority of the proposed system.

4.        The language and illustration layout in the text need to be improved, including uniform terminology usage and clear figure annotations.

5.        On page 3, “The FastSPM system presented in this work is composed of two units,  a control unit (CU) and a high - voltage unit (HVU).”  This paragraph mentions that the FastSPM system is based on a modular hardware architecture, but it is not detailed enough about the specific functions of the modules and how they work together.

6.        In terms of language, the use of unified terms in the whole paper, such as "Scanning probe microscopes (SPMs)", "these sensors", "the FastSPM system" and other expressions mixed in different locations, should be unified.

7.        The functions and effects of CU and HVU in Figure 2 should be explained in more detail in the figure notes. For example, the notes in Figure 2 can further illustrate how CU coordinates probe motion, data acquisition, and custom functions, and how HVU amplifies and adds the signal to the high-voltage offset signal of a standard SPM controller.

8.        In Section 2.2.4, the working principle of the resonance characterization tool and the process of determining the optimal frequency window in practice should be explained in more detail. For example: “By exciting an axis of choice (X, Y or Z) at distinct frequencies that are incremented step by step,  the amplitude and phase of the resulting local probe signal are measured,  providing a simple and highly sensitive access to the frequency - dependent scanning stage response along all three  axes. Optimal frequencies for the fast motion are those with the flattest frequency response, avoiding resonances,  which is typically best seen with the most sensitive excitation, i.e. Z [14].”  The process of determining the optimal frequency window is not described in detail here.

Comments on the Quality of English Language

In general, the English expression of the paper is more accurate and clear, which can effectively convey the core content of the research. The use of grammar is basically correct, and the choice of vocabulary is relatively appropriate, which can accurately describe related technologies and concepts.

However, there is still room for improvement in some aspects. For example, as mentioned above, the structure of some sentences is relatively complex, which may bring certain difficulties to readers' understanding. Appropriate simplification and adjustment of sentence structure can improve the readability of the article. In addition, the language style of the article is relatively simple, and some changes can be appropriately added to make the article more vivid and attractive.

Overall, the English quality of the article is qualified for academic expression, but it can be made even better with further optimization and improvement.

Author Response

Comments 1: In the introduction section, the author should add background information related to the rapid scanning probe microscope, such as the current common methods to improve scanning speed, the existing challenges, etc

Response 1: We thank the reviewer for pointing out a lacking state-of-the-art background. We addressed the issue by providing more details on the methods and techniques used by high-speed SPMs and adding the relevant literature. In particular, we expanded the paragraph in the introduction from line 38 to line 44.

Comments 2: The specific advantages of the FastSPM system's fast imaging function should be elaborated.

Response 2: Thanks to the reviewer comment we further highlighted the importance of fast SPM imaging by introducing some physical phenomena whose study would benefit from this technique from line 35 and extending the literature references with recent experimental results that used this technique from line 44.

Comments 3: More comparative experiments should be added to better demonstrate the superiority of the proposed system.

Response 3: The key advantage of our proposed architecture lies in its design as an FPGA-based addon, complementing rather than replacing existing commercial systems. This makes direct comparisons challenging, as no other system currently offers features like high-speed atom tracking in the same manner. While we acknowledge that additional comparative experiments could further highlight the superiority of our system, we believe that since this manuscript is intended for an FPGAoriented special issue, it is more appropriate to focus on the architectural innovations. We plan to present a detailed comparison of scientific results in a future publication. However, we fully understand the reviewer's point, and have included additional comments in lines 533–545 to address the performance and characteristics of other systems.

Comments 4: The language and illustration layout in the text need to be improved, including uniform terminology usage and clear figure annotations.

Response 4: In response to the reviewer's comment, we have improved the annotations of several figures, while also considering feedback from another reviewer who requested for more concise captions. To address both points, we’ve added reading aids within the text rather than in the captions. Additionally, we have standardized the terminology throughout the manuscript, particularly when distinguishing between SPM and FastSPM. We appreciate the reviewer's input, which has helped improve both the figure presentation and overall consistency.

Comments 5: On page 3, “The FastSPM system presented in this work is composed of two units, a control unit (CU) and a high - voltage unit (HVU).” This paragraph mentions that the FastSPM system is based on a modular hardware architecture, but it is not detailed enough about the specific functions of the modules and how they work together.

Response 5: We have revised and expanded the paragraph where the control unit (CU) and highvoltage unit (HVU) are introduced to better emphasize that the CU coordinates movement and data acquisition, while the HVU interfaces with the high-voltage signals. This introductory paragraph is intended to provide a brief overview of their functions, and the following section offers a more detailed explanation of their roles and how they work together. We believe this structure balances clarity with the need for in-depth discussion later in the manuscript.

Comments 6: In terms of language, the use of unified terms in the whole paper, such as "Scanning probe microscopes (SPMs)", "these sensors", "the FastSPM system" and other expressions mixed in different locations, should be unified.

Response 6: In an effort to unify and clarify terminology we modified the manuscript avoiding the use of “sensors” to identify the SPMs. Moreover, we better highlight that the FastSPM system is an add-on system applicable to standard SPMs commercially available.

Comments 7: The functions and effects of CU and HVU in Figure 2 should be explained in more detail in the figure notes. For example, the notes in Figure 2 can further illustrate how CU coordinates probe motion, data acquisition, and custom functions, and how HVU amplifies and adds the signal to the high-voltage offset signal of a standard SPM controller.

Response 7: We thank the reviewer for the helpful suggestion. To address both this comment and another reviewer's request to keep Figure 2's caption concise, we have carefully balanced the level of detail. We added some information to the figure caption, while further explaining the CU’s role in coordinating probe motion, data acquisition, and custom functions, as well as the HVU’s signal amplification, within the main text. This way, we ensure clarity without making the caption overly detailed.

Comments 8: In Section 2.2.4, the working principle of the resonance characterization tool and the process of determining the optimal frequency window in practice should be explained in more detail. For example: “By exciting an axis of choice (X, Y or Z) at distinct frequencies that are incremented step by step, the amplitude and phase of the resulting local probe signal are measured, providing a simple and highly sensitive access to the frequency - dependent scanning stage response along all three axes. Optimal frequencies for the fast motion are those with the flattest frequency response, avoiding resonances, which is typically best seen with the most sensitive excitation, i.e. Z [14].” The process of determining the optimal frequency window is not described in detail here.

Response 8: Following the reviewer suggestion we better detailed and made clearer the usage of the resonance characterization tool at lines 434--438. Also, a direct referencing to Dri et al. [https://doi.org/10.1016/j.ultramic.2019.05.010] is given for the details of the characterization procedure, including experimental results. In particular the sentence has been rephrased.

Reviewer 3 Report

Comments and Suggestions for Authors

I have read the paper with high academic interest. This work present a fully redesigned FPGA-based instrument that can be integrated into most commercially available, standard scanning probe microscopes. This instrument not only significantly accelerates the acquisition of atomic-scale images by orders of magnitude, but also enables the tracking of moving features. However, logic problem of structure. The work requires a careful revision, both technically and grammatically.

1.      In the last two paragraphs of the introduction, the introduction about the content of the article can be simplified.

2.      Figure caption for Figure 2. Figure caption only needs to give a brief introduction to the content of the picture, the detailed part needs to be introduced in the text

3.      Figure 10 should be placed in section 3 Experimental test measurements.

4.      References should be carefully revised for standardization and harmonization according to the journal guidelines for authors.

Comments on the Quality of English Language

I have read the paper with high academic interest. This work present a fully redesigned FPGA-based instrument that can be integrated into most commercially available, standard scanning probe microscopes. This instrument not only significantly accelerates the acquisition of atomic-scale images by orders of magnitude, but also enables the tracking of moving features. However, logic problem of structure. The work requires a careful revision, both technically and grammatically.

1.      In the last two paragraphs of the introduction, the introduction about the content of the article can be simplified.

2.      Figure caption for Figure 2. Figure caption only needs to give a brief introduction to the content of the picture, the detailed part needs to be introduced in the text

3.      Figure 10 should be placed in section 3 Experimental test measurements.

4.      References should be carefully revised for standardization and harmonization according to the journal guidelines for authors.

Author Response

Comments 1: In the last two paragraphs of the introduction, the introduction about the content of the article can be simplified.

Response 1: As pointed out by the reviewer, the last two paragraphs of the introduction were excessively verbose and for this reason they have now been shortened to improve readability.

Comments 2: Figure caption for Figure 2. Figure caption only needs to give a brief introduction to the content of the picture, the detailed part needs to be introduced in the text

Response 2: We thank the reviewer for the comment. Nevertheless, this comment is in contrast with another reviewer, which suggested adding some details in figures caption, and especially in figure 2. In an effort to accommodate both feedback we removed most of the details from the caption, better highlighting them in the text, but added some small notes based on other reviewer’s feedback.

Comments 3: Figure 10 should be placed in section 3 Experimental test measurements.

Response 3: While we recognize that Fig. 10 seems part of the Discussion and Conclusions section, this was due to Latex layout as the figure is included and referenced in Section 3.2. Nevertheless, we improved the layout of the revised manuscript to avoid possible confusion.

Comments 4: References should be carefully revised for standardization and harmonization according to the journal guidelines for authors.

Response 4: We thank the reviewer for their consideration and helpfulness in enhancing our work. We have corrected the references according to the “Reference List and Citations Style Guide for MDPI Journals”. These changes are not highlighted to avoid issues with latex compilation.

Reviewer 4 Report

Comments and Suggestions for Authors

The manuscript "Design of an FPGA-Based Controller for Fast Scanning Probe Microscopy" focuses on a further improvement of a cutting-edge imaging technique, which is scanning probe microscopy. The authors provide a solution for enhancing time resolution capabilities of this technique. The manuscript is interesting and has potential, it fits the scope of Sensors.

The submitted paper can be considered for publication provided that the following issues are addressed:

1. It was not quite clear from the text: did the authors use a standard commercial SPM equipped with a new time-resolution controller or a custom SPM unit?

2. Fig. 9 - what is the size scale of images?

3. Fig. 10 - was it necessary to add these results to the Discussion and Conclusions section? Maybe they fit better previous sections?

4. The readability of the manuscript can be enhanced if the authors briefly summarize them in a short separate Conclusions section (current Discussion and Conclusions section looks more like Discussion).

5. What is the time resolution threshold of the proposed technique?

6. Atom tracking produces a certain impression. Can the proposed technique be potentially used for tracking processes at a supramolecular scale?

Author Response

Comments 1: It was not quite clear from the text: did the authors use a standard commercial SPM equipped with a new time-resolution controller or a custom SPM unit?

Response 1: To better clarify this, we better highlight throughout the manuscript that the FastSPM system acts as an add-on controller for commercially available SPMs. The use of standard commercial SPMs with the FastSPM system is also reported in Section 3.1 and it’s the strength of the system, which does not need custom SPMs.

Comments 2: Fig. 9 - what is the size scale of images?

Response 2: We included the pixel resolution in the Figure caption and added a size scale on the images to improve readability.

Comments 3: Fig. 10 - was it necessary to add these results to the Discussion and Conclusions section? Maybe they fit better previous sections?

Response 3: While we recognize that Fig. 10 seems part of the Discussion and Conclusions section, this was due to Latex layout as the figure is included and referenced in Section 3.2. Nevertheless, we improved the layout of the revised manuscript to avoid possible confusion.

Comments 4: The readability of the manuscript can be enhanced if the authors briefly summarize them in a short separate Conclusions section (current Discussion and Conclusions section looks more like Discussion).

Response 4: We appreciate the reviewer's suggestion and have implemented it by renaming the "Discussion and Conclusions" section to "Discussion" and making some modifications to its content. Additionally, we have added a separate, brief Conclusions section to enhance the manuscript’s readability.

Comments 5: What is the time resolution threshold of the proposed technique?

Response 5: We thank the reviewer for the interesting question. The FastSTM system is capable of fast imaging up to 100 x 100 pixels at 200 frames/s (or 200 x 200 pixel @ 100 frames/s) and of particle tracking with a 5 us time resolution, which is now reported in lines 520--523. The time resolution of these techniques is limited by the HVU bandwidth of 20 kHz, now indicated in line 137.

Comments 6: Atom tracking produces a certain impression. Can the proposed technique be potentially used for tracking processes at a supramolecular scale?

Response 6: We thank the reviewer for the interesting question regarding this technique. For optimal tracking, the radius of the dithering movement during atom tracking should be comparable with the dimensions of the features to follow. This implies that, for bigger scales, the rotation speed should be reduced to avoid a non-linear response of the piezoelectric actuator, reducing, in turn, the available time resolution. Moreover, the topography of the feature plays an important role during the tracking and molecules with a complex-shape can be hard to track. As an example, Lechner et al [https://doi.org/10.1021/acs.jpcc.8b06866] were able to successfully track Pt clusters up to 19 atoms.

Reviewer 5 Report

Comments and Suggestions for Authors

Author Response

Comments 1: Regarding the electronics, it would be nice to have details on what HV supplies authors used to power the Apex opamps and what opamps were used on the Conditioning PCB.

Response 1: We thank the reviewer for the question. The Apex opamp are powered by an in-house designed AC/DC converter using LM337 and LM137 with a high voltage regulator layout. The conditioning PCB outputs uses OPA227s while the input uses an OPA2192.

Comments 2: The authors could comment on how the data are processed after being transferred to the host PC. The data are sampled at 25 MS/s, which gives 100 MB/s for two channels and 16 bit samples. Are these data processed in the real time? Are they used for preview of the scanned area?

Response 2: The maximum data throughput is actually 50 MB/s. This is because in fast-imaging mode only the signal input is sent to the host PC (In atom tracking mode, where both channels are used, they are averaged and downsampled to the atom tracking time-resolution, which is 5 us at best). This 50 MB/s data stream is processed in real time by the host PC software, which produces a first fast filtering and downsampling/interpolation to create the images. The processed data get saved in a HDF5 file and only a few frame/s are visualized in order to have a preview of the scanned area. All the save data can later be processed using the PyfastSPM python library for advanced processing and corrections.

Comments 3: Is non-contact scanning mode available? If yes, why authors used contact mode for scanning of the grating? The probe speed must have been on the order of mm/s, so I'm curious about probe wear during the measurements.

Response 3: Unfortunately, the non-contact mode is not available at the moment but we are currently working on it. As the reviewer correctly noted, the contact mode at these speeds is stressful for both the probe and the sample. As a matter of fact, in the AFMs measurements in Fig. 9 there are some triangular shaped artifacts in the background, which are typical of damaged AFM tips. Nevertheless, we were able to take different measurements with the same probe, which lasted for a few hours. On an interesting note, when used with STMs the FastSPM system sometimes improves the probe; we think that the fast movements are able to “shake off” impurities from it.

Comments 4: Please provide more details on the scans presented in Fig. 9. What was the pixel resolution in all cases? What grating exactly was used, what was the height of the profile? What cantilever was used? The NSC36 probe features 3 cantilevers with different stiffnesses...

Response 4: The pixel resolution in Fig. 9a and 9c is 100x100 pixels while in Fig. 9b is 120x120 pixels. We thank the reviewer for pointing this out and we have included this information in the revised manuscript in Fig. 9 caption. The grating used was an old (discontinued) TGX1 from Mikromasch with a 3 um pitch chess pattern and a 130 nm height. The cantilever used was the softer one, with a force constant of 0.6 N/m.

Comments 5: What is the origin of the oscillations that can be seen in the images in the first row in Fig. 9? What measurement mode was used in this case?

Response 5: The oscillations seen in Fig 9 (which now has changed to accommodate the requests of the other reviewers) are caused by the interaction of the tip with the sample. In fact, the bandwidth of the Z-feedback of the controller is not sufficient to follow the fast-scanning movement, hence sometimes it is possible to see some collisions when large protrusions are present onto the surface. In a similar way, some images will display shaded areas like in Fig 9a. These are caused by the response of Z-feedback of the native controller which tries to follow the Pt5 cluster.

Comments 6: As the authors offer the module commercially, it would be nice to have at least a rough estimate on the price.

Response 6: While the authors are aware that the FastSPM system is commercially available from the Industrial Liaison Office of Elettra Sincrotrone Trieste, we developed the electronics described in this work for internal use and scientific purposes and are not involved in the commercialization process. For this reason, the authors are unable to provide the requested information.